# Competencies in the Robotics of Care for Nursing Robotics: A Scoping Review

**DOI:** 10.3390/healthcare12060617

**Published:** 2024-03-08

**Authors:** Blanca Gonzalo de Diego, Alexandra González Aguña, Marta Fernández Batalla, Sara Herrero Jaén, Andrea Sierra Ortega, Roberto Barchino Plata, María Lourdes Jiménez Rodríguez, José María Santamaría García

**Affiliations:** 1Meco Health Centre, Community of Madrid Health Service (SERMAS), 28880 Madrid, Spain; 2Research Group MISKC, Department of Computer Science, University of Alcala, University Campus, 28805 Alcala de Henares, Spainlou.jimenez@uah.es (M.L.J.R.); 3Santa Cristina University Hospital, Community of Madrid Health Service (SERMAS), 28009 Madrid, Spain; 4Mejorada del Campo Health Centre, Community of Madrid Health Service (SERMAS), 28840 Madrid, Spain; 5Computer Science Department, University of Alcala, 28805 Madrid, Spain

**Keywords:** robotics, nursing informatics, nursing theory, professional competence, clinical competence

## Abstract

In parallel with the development and design of different technological advances, competencies in nursing have advanced. With the development of robotics, it is expected that nursing robotic competencies will also increase. The aim of this study is to review the competencies in nursing robotics. A review was conducted between January 2017 and December 2023. The search strategy was carried out in the MEDLINE database (through PubMed). This review explores the developmental competencies in nursing robotics and informatics. The data extraction in this review included an intentional search for competencies and learning outcomes in engineering and robotic programs. A total of 340 competencies and program outcomes were reviewed. The synthesis of the data established a total of 17 developmental competencies in nursing robotics based on this knowledge extraction, which we organized into five categories: assessment, diagnosis, planning, intervention (implementation) and evaluation. This review suggests that nursing robotic competencies for the development of care robotics are still scarce, and there is an opportunity for the development of competencies and the definition of new roles in the area of nursing informatics in order to adapt to the new health care demands of society.

## 1. Introduction

The World Health Organization emphasizes innovation and technological progress as key to the challenges in our society among its Sustainable Development Goals. Information and knowledge technologies are at the forefront of the response to COVID-19. Therefore, harnessing these digital technologies and their potential for accelerating human progress, bridging the digital divide and developing knowledge societies are all called for [1,2].

The digital transformation of health care can improve health outcomes by enhancing, among other things, self-care and person-centered care, as well as expanding professional knowledge, skills and competencies in health service delivery [2].

The Global Strategy on Digital Health 2020–2025 states that digital health is “*the field of knowledge and practice associated with the development and use of digital technologies to improve health*”. It encompasses other uses of digital technologies, such as the internet of things, advanced computing, artificial intelligence and robotics, among other things, in health [2]. 

Similarly, the 2030 Agenda for Sustainable Development stresses the importance of expanding the competencies of health care professionals in these settings, so that these professionals are prepared to implement or use technology. Building these competencies involves the transmission of knowledge and skills, as well as the attitude of the professionals themselves. Strategic Objective 4 mentions that these competencies should be included in the curriculum and training of health professionals as well as in the different levels of education and training. These competencies range from computer science, strategic planning, finance and management to health sciences and care delivery [2].

Harnessing these technologies and supporting their sustainable development raises several issues in terms of digital skills and competencies among segments of society, as future jobs will require information and communications technology (ICT) skills [3]. 

The Organization for Economic Cooperation and Development (OECD) highlights that the future of work will require technical and professional skills, sometimes sector-specific, such as the operation of robots [3].

Robotics and artificial intelligence are technologies that will have a significant impact on the future of our society. This has led organizations such as the European Parliament to develop a framework of the ethical aspects of artificial intelligence, robotics and related technologies and includes their use as high-risk when applied to health care in terms of the possibility that their use could lead to a violation of fundamental rights and safety standards [4].

Robotics is defined as “the technique of applying computer science to the design and use of devices that, in place of humans, perform operations or work, typically in industrial facilities” by the *Dictionary of the Spanish Language* (Royal Spanish Academy). The Robotics and Automation Society of the Institute of Electronics and Electrical Engineers states, “Robotics focuses on systems incorporating sensors and actuators that operate autonomously or semi-autonomously in cooperation with humans. Robotics research emphasizes intelligence and adaptability to cope with unstructured environments” [5]. The International Standard Organization (ISO) 8373:2021 defines the term robot as a “programmed actuated mechanism with a degree of autonomy to perform locomotion, manipulation or positioning” [6].

Robotics are developed, deployed and used by human beings. This implies that the choices humans make will determine the impact of robotics on society [4]. 

Some authors underline that in recent years, the use of robotics has spread to other areas, including health sciences and, thus, nursing [5].

We found some examples of robots used to facilitate various care tasks in the context of improving the independence of elderly people [7,8]. Examples include robots that assist with tasks involving feeding, safety, environmental hygiene and cleaning and companionship [9,10]. 

The latest revision of the International Council of Nurses (ICN) Code of Ethics for Nurses 2021 states, “Nurses ensure that the use of technology and scientific advances are compatible with the safety, dignity and rights of people. In the case of artificial intelligence or devices, such as care robots or drones, nurses ensure that care remains person-centered and that such devices support and do not replace human relationships”. The Code includes among the applications of the elements for nurse educators and researchers the following: “seek opportunities to evaluate the short and long-term ethical consequences of the use of diverse technologies and emerging practices, including innovative equipment, robotics, genetics and genomics, stem cell technologies and organ donation”. This is the first mention of robotics in this document [11].

The research work in robotics has just started. This research area could be addressed by nursing informatics. From this perspective of nursing informatics, it has become necessary to incorporate the vision of health care [12]. 

In this regard, according to robotics, the American Nurses Association (ANA) states that nursing is one of the professions that will be most impacted by the use of robotics and has reviewed the service robots that currently work to support nurses [5]. 

In addition, it is essential that nurses are involved in the initial design of systems relating to health care [13].

Thus, in parallel to the development and design of the different technological advances, competencies in nursing informatics have been advanced [14].

Traditionally, competencies in nursing informatics have been classified into three domains: basic computer skills, informatics knowledge and informatic skills [15]. 

Those competencies were structured into four levels depending on the level of nurse: beginning nursing, experienced nurse, informatics specialist and informatics innovator [16]. This has been followed by other alternatives such as Health Information Technology Competencies (HITCOMP) [17], a tool to assess health information technology skills, or Technology Informatics Guiding Education Reform (TIGER), an initiative to provide informatics/e-health tools and resources to health care workers [18]. 

Therefore, technological education in nursing is changing from an approach based on the use of technologies to an approach based on the design and construction of the devices itself. For that reason, the following question arises, what are the nursing competencies in nursing robotics? To this end, this article focuses on identifying what the nursing competencies are in nursing robotics. Therefore, the aim of this study is to review the competencies in nursing robotics. 

## 2. Materials and Methods

### 2.1. Design

A scoping review of the competencies in nursing robotics based on the PRISMA statement [19] was used in this review. 

### 2.2. Search Methods

The search strategy was conducted in the MEDLINE database using the search engine PubMed and included articles published between January 2017 and December 2023. The year 2017 was chosen as the starting date because in that year, the European Parliament published the document “*Civil law rules on robotics”*, included in the resolution “*Framework of ethical aspects of artificial intelligence, robotics and related technologies*” [4], which would mark an important institutional milestone for the development of this field of research.

The construction of the search strategy was carried out by six researchers, nurses with expertise in the fields of health and computer science research. The literature search in PubMed and the data analysis was conducted by the main research.

To construct the search strategy, a combination of medical subject heading (MeSH) terms and keywords were used:Selection of search terms about nursing, robotics and competencies: A literature search on the three core elements of the research: nursing, robotics and competencies. For the search, a series of keywords in MeSH terminology and natural language using truncation.Selection of search terms about nursing, informatics and competencies: A literature search on the three core elements of the research: nursing, informatics and competencies. For the search, a series of keywords in MeSH terminology and natural language using truncation. In this case, the search for the term informatics was restricted to the single use of the term MESH and not to the natural language.Identification of keywords to distinguish competencies of use and competencies of development: A distinction was made between those that were related to competencies of use and competencies of development. For this purpose, the *Dictionary of the Spanish Language* (Royal Spanish Academy) was used to extract the definitions of the use and development of the concepts as well as an electronic resource for the search of synonyms based on the Larousse dictionary. The translations were carried out using the *Oxford English Dictionary*.

The keywords identified to describe the search strategy are shown in Table 1. All of the selected search terms were used for the search strategy.

Once the keywords were selected, four categories were established to separate the records: (1)Developmental and non-use competencies.(2)Use and developmental competencies.(3)Neither use nor developmental competencies.(4)Use and non-developmental competencies.

The search strategy selected the records which were those included in the first category: developmental and non-use competencies. The search strategy included the language filters (only English and Spanish were admitted) and the time limit mentioned above. The search strategy included the Boolean operators “OR”, “AND” and “NOT” (see search strategy in Table 2). 

However, some articles prior to the date of inclusion for this study were reviewed because of their interest for the topic under study. All of them were referenced in some of the articles included in the review.

### 2.3. Data Extraction

The data from each of the articles selected were extracted using a table, which contained the following elements: bibliographic elements (authors, year of publication, title, country), type of article, research objective, thematic classification according to whether it addresses nursing robotic competencies and/or informatic nursing competencies relating to: level of education targeted (undergraduate, postgraduate), care setting (hospital setting, primary care setting) or whether it is theoretical, leadership or the future of all of these. This analysis was carried out by the principal investigator of this study (Appendix A).

For the proposal of the nursing competencies development in care robotics, a last phase was carried out in January 2024. A knowledge extraction of the results of the review was carried out for nursing informatic competencies and an intentional knowledge extraction was carried out for engineering and robotic competencies. 

For the nursing informatic competencies section, the results of the review were taken into account (Appendix A):The 11 thematic classifications of competencies established by Strudwick [20].The classification of the 10 core domains of health informatics proposed by the Health Informatics Core Competencies of the American Medical Informatics Association (AMIA) [21] which used the terms knowledge, skills and attitudes (KSAs) for the categorization of competencies and were identified by the following acronyms: “K, S, A + F + number”.The classification of competencies in informatics nursing proposed by Staggers [16], in which the competencies were categorized into four levels (beginner nurse, experienced nurse, informatics specialist and informatics innovator) with three main categories (computer skills, informatics knowledge and informatic skills). A total of 212 competencies were selected for the present study corresponding to the categories of informatics specialist and informatics innovator. The competencies were identified and coded by the lead author for further classification. The following acronyms were used for coding: “IS + number” (informatics specialist) and “II + number” (informatics innovator).

For the robotics engineering section, an intentional search was carried out in Spanish and European study programs related to engineering and robotic competencies. The following were taken into account (Appendix A):A total of 25 learning objectives from the EUR-ACE^®^ framework standards and guidelines [22], classified according to whether they were aimed at Bachelor or Master degree programmes in engineering and oriented towards 8 learning areas: knowledge and understanding; engineering analysis; engineering design; investigations; engineering practice; making judgements; communication and team-working and lifelong learning. The 25 learning objectives were coded and identified by the lead author with the acronyms “B + number” (Bachelor Degree Programs) and “M + number” (Master Degree Programs).A total of four teaching plans (83 learning outcomes) included in the White Paper on Robotics [23] published in 2011 in Spain and the Curriculum in Robotics Engineering published in 2022 [24].

Data were extracted resulting in the proposed development of nursing robotic competencies in care robotics.

The synthesis of the data was carried out by the main researcher. The competencies for nursing development in care robotics were grouped into five categories corresponding to the five phases of the nursing process: (1) assessment, (2) diagnosis, (3) planning, (4) intervention (in this case implementation) and (5) evaluation. These phases have been used by Sarrión-Bravo et al. [25] to define competencies in spiritual and emotional care, and by García-Día [26] to relate them to product life cycle and project process groups in the book *Project Management in Nursing Informatics.*

## 3. Results

A total of 21 records were obtained from the search strategy about nursing, robotics and competencies. A total of 88 records were obtained from the search strategy about nursing, informatics and competencies. The 109 records were reviewed by reading the title and abstract. Finally, 16 articles were selected for inclusion in the descriptive review. Figure 1 presents the flow chart of the search summary. The Appendix A summarizes the data extracted according to the elements described in Section 2.3, Data Extraction, of the Materials and Methods section. However, the results obtained from the literature review are described below.

### 3.1. Literature Review of Competencies of Development in Nursing Robotics

Salzmann-Erikson and Eriksson [27] have stated that there is research on attitudes, implementation in care settings as well as the evaluation of care robots, specifically with regard to care in the elderly. They called for the involvement of nursing in the management of the process through participation in the planning and design as well as the development of robotic-related competencies in the care team. To this end, they also called for the development of a new infrastructure, including in education, to respond to these new demands arising from the advancement of technology such as robotics.

### 3.2. Literature Review of Competencies of Development in Nursing Informatics

Some authors have considered a conceptual framework integrating interdisciplinary education in engineering and health informatics nursing. In their case, they have opted for a theoretical framework based on the TIGER initiative (structured learning based on competencies for all nurses, for beginner and experienced nurses) and the section on informatics proposed by the Quality and Safety Education for Nurses (QSEN), which organizes competencies based on the axes of knowledge, skill and attitude. In addition, it describes two instructor profiles for this interdisciplinary education [28]:An engineer who has completed a post-doctoral training in health care.A nurse who has earned a dual graduate degree in nursing and biomedical engineering.

All this combined with a learning method based on “Learn by doing” [28].

Another example is the eHealth4all@eu Pipeline course which follows the TIGER recommendations [29].

Regarding the design of technologies, some authors have highlighted an opportunity for nurse researchers to design technological devices in the field of study [30].

The theoretical framework proposed for the description of nurses with nursing informatic competencies focuses on three interrelated aspects: transformational leadership (morale modelling, vision stimulation, individualized consideration, charisma and idealized influence), nursing informatic competencies (nursing informatics awareness, computer operation competency, computer software management competency, nursing information operation competency, nursing information management competency) and innovation self-efficacy [31].

Competencies in nursing informatics:

Studies by Strudwick et al. showed the development of a learning plan based on a selection of competencies which demonstrated that training in them had increased knowledge, experience and confidence [20].

Monsen et al. compared the American Association of Critical-Care Nurses (AACN) Informatics and Healthcare Technologies Essential V and the AMIA Health Informatics Core Competencies [21].

Lozada-Perezmitre et al. focused on the development and validation of the Self-Assessment of Nursing Informatics Competencies Scale (SANICS) translated into Spanish [32]. The same scale was used by Choi et al. [33] and by Galacio [34] to assess nursing informatic competencies. 

Liu and Aungsuroch’s revision established 11 components: 1. personal traits; 2. professional clinical practice; 3. legal and ethical practice; 4. ensure safety and quality; 5. communication; 6. management of nursing care; 7. leadership; 8. cooperation and therapeutics practice; 9. teaching—coaching; 10. critical thinking and innovation; 11. professional development [35].

Other studies have measured competencies in nursing informatics among graduates of different years [36] while other authors have reviewed different curricula in order to explore curricula that have included specific courses on health informatics, nursing informatics or information and communication technologies for nursing care [37].

Chen et al. established general nurse, information nurse and information expert levels for learning nursing informatic competencies [38].

Other authors have reviewed the literature on nursing informatics education among nursing undergraduates [39].

Borycki et al. [40] used the work of Staggers et al. [16] for the mapping of informatic nursing competencies and adapted it to the development of these competencies in the field of patient safety.

### 3.3. Proposed Competencies for Nursing Development in Care Robotics

The result is the establishment of a total of 17 competencies for nursing development in care robotics. The results are shown in Table 3. The competencies are organized into five categories (first column): (1) assessment; (2) diagnosis; (3) identification of objectives; (4) intervention (in this case implementation) and (5) evaluation [25,26].

Second column indicates an identifier number (ID) created to identify each competency.

Third column indicates the text of the competencies.

Fourth column shows the relationship between each of the competencies with the competence themes established by Strudwick [20].

Each of the competencies has been proposed based on the related competencies and learning outcomes based on the knowledge extraction of the nursing informatics and robotics engineering competencies from the AMIA Health Informatics Core Competencies [21], the 212 nursing informatics competencies proposed by Staggers [16] for the informatics specialist and informatics innovator levels, the 25 learning objectives from the EUR-ACE^®^ framework standards and guidelines [22] and the 83 competencies and learning outcomes included in the selected curriculum [23,24]. 

Examples of these competencies and learning outcomes for each of the competencies in nursing robotics are shown in Appendix A.

## 4. Discussion

The aim of this article was to review and identify competencies in nursing robotics.

The advancement of technologies such as robotics has led to an analysis from a nursing perspective [5].

There are proposals that establish the competencies to be acquired by the different professions and citizens in the field of technology in a global manner, such as the DigComp strategy, which includes competencies related to robotics [41].

With regard to the results obtained from the search in the first phase, it should be stressed that most of them are oriented towards usability [42], acceptability [43,44] and ethical implications [45] among professionals. In this sense, Stokes and Palmer [46] have claimed that AI or AI carebot activities must meet three conditions: “first, it cannot transgress the core values of nursing—i.e., caring. Second, it cannot usurp important aspects of caring that can only meaningfully be carried out by human beings. Third, it must support, expand or improve opportunities for nurses to provide the uniquely human aspects of care”. Others have focused on a specific aspect of robotic development in the field of surgery [47].

Among the competencies not approved by Staggers et al. in 2002 [16] was “*writes an original computer program and modifies it*”. In addition, the competency for the computer specialist level “*discusses concepts and uses of robotics*” was approved but did not gain agreement to be determined as a valid competency. Both competencies, discarded at the time, have a valid content today and could be adopted for this study.

The choice of learning outcomes proposed by the European Network for Accreditation of Engineering Education standardizes them for a wide range of countries, as well as being an international seal of international quality [22]. However, there are other accreditations, such as ABET [48] and ARCU SUR [49].

In 2018, Glasgow et al. reviewed a number of training programs linking engineering and nursing to define the role of the nurse engineer [50]. In addition, recent studies have applied the paradigm of care robotics to the development of applications with this perspective [51].

With regard to the way of organizing the competencies identified and extracted, a classification has been used based on the phases in the nursing care process as opposed to the classification formats used for competencies in computerized nursing and which have served as a starting point for the development of this study.

The competencies listed could be included among the different classifications of informatics nursing as specific content adapted to the development of nursing robotics within the framework of informatics nursing.

### Limitations

This review may include potential limitations. First, only publications in English and Spanish were included. The authors only searched one database (PubMed), which may have excluded potentially important sources of information, and certain information may have been disregarded. There was a focus on the developmental competencies and as a result, this excluded papers about the use of competencies. Furthermore, in the future, the methodology could include a phase of content validity through experts.

## 5. Conclusions

This review is the first to investigate competencies in nursing robotics. The review shows that nursing robotics for the development of care robotics are still scarcely explored. Nursing informatics is a field with extensive studies in the literature about competencies. Nursing informatic competencies are part of the nursing informatics curriculum, but they are also the basis for the development of nursing robotics. In addition, the competencies and program outcomes of the engineering and robotics curriculum serve to broaden the competencies for the development of nursing robotics.

Just as competencies in nursing informatics have been advancing in parallel to the development of different technological advances, the inclusion of disruptive technologies such as artificial intelligence and robotics poses new challenges for nursing that must be faced from an approach based on training which focuses on the development of robotics in nursing.

All this exponential growth has opened up new fields of development for nursing. Among them is the profile of the robotic nurse, an expert in care robotics. Care robotics is a type of robotics oriented towards human care. Therefore, this profile has as its object of research robotics oriented to human care under the paradigm of human care knowledge models and the conceptual framework of nursing informatics. This synthesis of competencies proposes a model of construction of devices from the paradigm of robotics of care assuming the precepts of the scientific nursing discipline.

## Figures and Tables

**Figure 1 healthcare-12-00617-f001:**
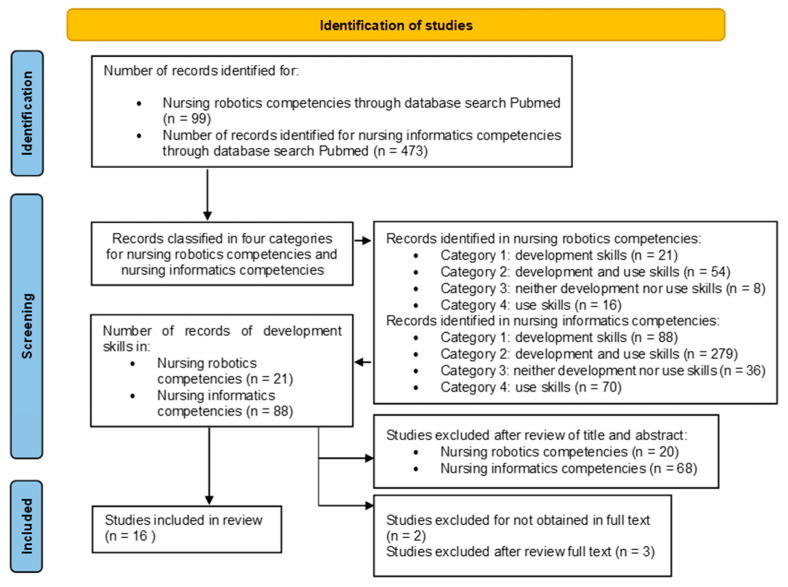
PRISMA flowchart of descriptive review process (January 2017 to December 2023).

**Table 1 healthcare-12-00617-t001:** Search terms selected.

Areas	Search Terms (MeSH)	Search Terms (Natural Language)
Nursing	Nursing, Practical	Nurs*
	Nursing	
	Nursing Care	
	Nursing Services	
	Nursing Faculty Practice	
	Nursing Informatics	
	Nurses	
	Nursing, Practical	
	Nursing	
	Nursing Care	
Robotics	Robotics	Robot*
Informatics	Informatics	-
Competences	Professional competence	Competenc*
	Clinical competence	Skill*
		Abilit*
**Type of Competence**	**Key Word**	**Search Terms (Natural Language)**
Competence of use	Use (using); use of	use*
	Utilization	utiliz*
	Application	empl* appl*
	Handle; management	management* handl*
	Practice	practis* practic*
	Use (using); use of	use*
	Utilization	utiliz*
	Application	empl* appl*
Development competence	Development (developing)	develop*
	Growth (growing)	grow*
	Increase (increasing)	increas*
	Progress, improvement	progress* improv*
	Advance	advanc*
	Evolution	evolut*

The use of asterisk means natural language using truncation.

**Table 2 healthcare-12-00617-t002:** Search strategy.

Search strategy for competencies of development in nursing robotics	(((nursing, practical[MeSH Terms]) OR (nursing[MeSH Terms]) OR (nursing care[MeSH Terms]) OR (nursing services[MeSH Terms]) OR (nursing faculty practice[MeSH Terms]) OR (nursing informatics[MeSH Terms]) OR (nurses[MeSH Terms]) OR (nurs*[Title/Abstract])) AND ((robotics[MeSH Terms]) OR (robot*[Title/Abstract])) AND ((professional competence[MeSH Terms]) OR (clinical competence[MeSH Terms]) OR (competenc*[Title/Abstract]) OR (skill*[Title/Abstract]) OR (abilit*[Title/Abstract])) AND ((develop*[Title/Abstract]) OR (grow*[Title/Abstract]) OR (increas*[Title/Abstract]) OR (progress*[Title/Abstract]) OR (improv*[Title/Abstract]) OR (advanc*[Title/Abstract]) OR (evolut*[Title/Abstract])) NOT ((use*[Title/Abstract]) OR (utiliz*[Title/Abstract]) OR (empl*[Title/Abstract]) OR (appl*[Title/Abstract]) OR (management*[Title/Abstract]) OR (handl*[Title/Abstract]) OR (practis*[Title/Abstract]) OR (practic*[Title/Abstract]))) AND (english[Filter] OR spanish[Filter]) AND (2017:2023[pdat])
Search strategy for competencies of development in nursing informatics	(((nursing, practical[MeSH Terms]) OR (nursing[MeSH Terms]) OR (nursing care[MeSH Terms]) OR (nursing services[MeSH Terms]) OR (nursing faculty practice[MeSH Terms]) OR (nursing informatics[MeSH Terms]) OR (nurses[MeSH Terms]) OR (nurs*[Title/Abstract])) AND (informatics[MeSH Terms]) AND ((professional competence[MeSH Terms]) OR (clinical competence[MeSH Terms]) OR (competenc*[Title/Abstract]) OR (skill*[Title/Abstract]) OR (abilit*[Title/Abstract])) AND ((develop*[Title/Abstract]) OR (grow*[Title/Abstract]) OR (increas*[Title/Abstract]) OR (progress*[Title/Abstract]) OR (improv*[Title/Abstract]) OR (advanc*[Title/Abstract]) OR (evolut*[Title/Abstract])) NOT ((use*[Title/Abstract]) OR (utiliz*[Title/Abstract]) OR (empl*[Title/Abstract]) OR (appl*[Title/Abstract]) OR (management*[Title/Abstract]) OR (handl*[Title/Abstract]) OR (practis*[Title/Abstract]) OR (practic*[Title/Abstract]))) AND (english[Filter] OR spanish[Filter]) AND (2017:2023[pdat])

The use of asterisk means natural language using truncation.

**Table 3 healthcare-12-00617-t003:** Competencies in nursing robotics (see bibliographic examples in Appendix A).

Categories [25,26]	ID	List of Competences	Competency Theme [20]
Assessment	1	Develop a new conceptual framework for the development of robotics of care.	1
	2	Demonstrate knowledge of: Care and health concepts for integration into robotic systems. Mathematics, computer science, electronics and other basic sciences for the development of health care-oriented robotic systems. The evolution of robotics throughout history. Current trends in robotics in relation to the field of health care.	1
	3	Discriminate digital information resources, differentiate trends in theories of care, acquire the ability to follow the evolution of care robotics and identify the different existing research methodologies for research and evaluation of the impact of robotics at the health level from a bioethical perspective.	1, 4, 5, 7
	4	To be aware of current developments in intellectual property and patents, the ethics and regulations. Define/discuss ethical principles and responsibilities through consultation and application of codes of practice and safety regulations in the field of robotics and health care. Ability to apply and incorporate laws and regulations of robotics practice in the health care setting.	5, 6
Diagnosis	5	Ability to: Apply knowledge derived from conceptual models in the substantiation of care issues taking into account current issues of vulnerability and fragility. Develop critical thinking and new theoretical and practical proposals in the field of care robotics and applied research in this field. Demonstrate consideration of the advantages and limitations in the use of robotics in health care as well as individual needs and their context.	1, 4
	6	Analyze through logical–mathematical reasoning the representation and formalization of health problems for their implementation in robotic systems and demonstrate the capacity and ability to generate logical response algorithms to a given problem for subsequent implementation in robotic systems.	1, 9
	7	Combining knowledge management technologies for information gathering in digital health environments for the development of robotic care systems. Demonstrate the capacity and ability to use computer resources for the analysis and processing of data that allows its interpretation in different care situations for its subsequent use in the development of robotic systems. Demonstrate and infer knowledge of the various existing (standardized) health and/or health-related taxonomies and professional languages for use in the development of ontologies in robotic systems of care.	1, 2
	8	Design new applications and creative resources in the field of robotics applied to health care.	3, 8, 9
Planning	9	Assess available robotic tools according to their problem-solving capabilities in health care and acquire programming competencies and skills in handling hardware elements with an approach based on a person-centered knowledge model applied in care robotics.	1, 2, 8, 9, 10
	10	Demonstrate knowledge and acquire skill in the: Design of drawings and wiring diagrams of hardware elements within a health care-based developmental framework. Development and design of robotic systems oriented towards health care in compliance with the established technical requirements.	2, 8, 9, 10
	11	Acquire ability to: Develop and design complex products (artefacts, devices, in this case robotics) in the field of study of health sciences to meet established requirements including awareness of non-technical aspects such as society, health and safety, environment, economy and industry; select and apply relevant methodological designs. Select and apply relevant methodological designs or use creativity to generate new and original design methodologies with a health care-oriented approach.	1, 3, 4, 5,6, 7, 11
Implementation (intervention)	12	Demonstrate knowledge of materials, equipment, tools and technologies and their limitations for the development of robotic devices.	1, 2, 8,9, 10
	13	Acquire: Practical skills, including the use of computer tools, to solve complex problems related to care robotics. Capacity and skills in the construction and assembly of prototypes of robotic care systems. Knowledge about the verification and validation of robotic prototypes of care.	1, 2, 8, 9, 10
	14	Design strategies to involve clinicians in the design, selection, implementation and evaluation of health care applications and systems.	8, 9, 10, 11
Evaluation	15	Acquire the ability to: Manage technical or professional activities and projects in the field of care robotics. Recognize the need for independent lifelong learning about care robotics and to take part in it. Develop research to examine the impact and determine further needs for the application of robotics in nursing.	3, 4, 11
	16	To be able to: Make effective oral or written communications in order to be able to disseminate both research projects and results in care robotics in different scientific environments. Communicate findings, knowledge and rationale underpinning studies and research related to care robotics to specialist and non-specialist audiences in a clear and unambiguous way, and using suitable language for multidisciplinary audiences.	3, 4, 11
	17	Recommend policies and procedures to improve the practices or objectives of health care-oriented robotic systems.	11

## Data Availability

The data that support the findings of this study are available in the Appendix A and on request from the corresponding author.

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
