# Peer review of "Competencies in the Robotics of Care for Nursing Robotics: A Scoping Review"

_healthcare, 2024, doi:10.3390/healthcare12060617_

Round 1

Reviewer 1 Report

Comments and Suggestions for Authors

Dear Authors,

Thank you for the opportunity to read your work. I find your manuscript very valuable. You raise an important issue, an issue that is beginning to penetrate more and more intensively into nursing. The development of new technologies, increasing digitization, and the use of AI are the near future of nursing. Nurses must face these challenges and cannot stand by them indifferently. I'm glad you touched on this topic.

The methodology is thoroughly developed and the results are presented in an interesting way. It's a pity that you only looked at PubMed, but I understand that this was due to some limitations beyond your control.

Author Response

Thank you for taking the time to review this manuscript. Please see the attachment. 

Reviewer 2 Report

Comments and Suggestions for Authors

Relevant article and very interesting topic. It must be published.

Reinforcement of the soldiers found is suggested.

In the method, justify the existence of only one of the authors as an evaluator of the work.

in table 3 it should be more explicit what the item in the second column relates to. Possibly repeat or place in-line divisions of the tables according to the group.

In the discussion and conclusion, it would be important to have more emphasis on the use of the results of this review, integrating them as intended. The conclusion can also reinforce the benefits of systematization.

Comments on the Quality of English Language

On line 90 there is an extra 6 in the text.

Author Response

(The authors gave the same response as above.)

Reviewer 3 Report

Comments and Suggestions for Authors

While this review attempts to delve into competencies in nursing robotics, its methodology raises concerns. The narrow focus on PubMed as the primary database for literature search may have limited the breadth of studies considered, potentially overlooking relevant research from other sources. Furthermore, the synthesis of data from different phases appears disjointed, with the transition between phases not clearly elucidated. Despite claiming to identify 17 key competencies, the review fails to provide a robust analysis of how these competencies were derived or validated. Moreover, the assertion that nursing robotics competencies are lacking and the call for new roles in nursing informatics seems speculative, lacking substantial evidence or empirical support. Overall, this review falls short of providing a comprehensive understanding of competencies in nursing robotics and fails to offer actionable insights for advancing the field.

Author Response

(The authors gave the same response as above.)

Reviewer 4 Report

Comments and Suggestions for Authors

Dear authors, thank you very much for the opportunity to read your manuscript, which addresses a review on an interesting topic regarding nurses' competencies in robotics.

As this is a review study, my comments will be made in accordance with the PRISMA guidelines.

Has the research protocol been registered in PROSPERO or similar? This is important for many subsequent comments on the developed methodology.

In the material and methods section, the authors state that this is a systematic review; however, in the title they only state that it is a review. This criterion should be unified.

Keywords: The keywords are adjusted to MeSH terms, except "Competency" which does not correspond to MeSH; in my opinion "Competency" is a non-specific term ( you can use an alternative term such as "Clinical Competence" or "Competency-Based Education", if you consider it fits).

Abstract: In my opinión, the abstract should be better adjusted to PRISMA (improve the information on the method: inclusion/exclusion criteria, methods to assess risk of bias in the included studies, ). There is redundancy in the following information: "The aim of this study is to review competencies in nursing robotics ... This review explores the development competencies in nursing robotics and informatics".

Introduction: The introduction is well described.

Review in the first paragraph ¿COVID-19.1?

Check terms with redundancies in the introduction (e.g: "potential" in the first paragaph).

Review the use of parentheses in the expression: "The latest revision of the (International Council of Nurses (ICN)". (page 2, line 80).

Explain this content and acronyms in more detail: "This has been followed by other alternatives such as HITCOMP [17] or TIGER [18] (page 3, line 104-5).

In relation to the aim, it is not necessary to specify a new heading at the end of the introduction. Just describe the aim. Also, you should not point out the method in this section. It would be convenient to end the introduction with the research question guiding the review and finally describe the aim.

Methods:

In the methods section I think you should simplify the description of the systematic review process. PRISMA does not point out the existence of phases in the review (phase 1 including subphase 1 and subphase 2; phase 2 including subphase 1 and subphase 2; searches in different subphases and in different phases). The different phases do not correspond since the review process is perfectly described in the methodology according to PRISMA: design, information sources, search strategy, selection process, data collection process, study assessment.

Finally you describe a phase 3 to propose competences in care robotics (from my point of view this phase is the results of the review and does not correspond to any phase).

Desing: should be drafted in a simpler way in this section.

Search methods: Have you only searched MEDLINE (PubMed is the search engine and MEDLINE the database), because you have not searched other major databases: WOS, SCOPUS, CINAHL at least? You should describe the terms used (MeSH and free term), as well as the Boolean operators used in the searches. Remove the heading "Phase 1" in search methods.

Inclusion criteria statement should be included for this information: "The result of this search with the language filters (only English and Spanish were admitted) and the time limit mentioned above gave a total of 99 articles" (page 4, line 138). 

Do not include in methods the information on the number of records retrieved (n=99 are not articles, they are retrieved records). This information will be included in the flowchart in the results section (as described in Figure 1). 

flowchart: The flowchart should go in results. In my opinion, the flowchart is not correctly elaborated. The fact that the search strategies are different (using two strategies: robotics competences vs. informatics competences) does not imply that the process flow in the diagram has two different paths. The number of records identified in PubMed is the sum of all the search strategies (duplicate records will have to be eliminated in each of the searches (or databases if other sources were used)). It is also not correct to identify in the screening phase the records that correspond to each established category (this information will be the results of the review, not of the screening process). The diagram does not include information on elimination in the screening phase by title and abstract (inclusion criteria...) and eliminated in the full-text phase (critical appraisal and inclusion criteria).

Separating the results into 4 categories does not correspond to the search process. This categorization corresponds to the method of synthesis of results. It is described in methods, but they are organized in results. The identification, screening and inclusion process should not distinguish categories of results since the studies are not yet included (a different thing is that they establish exclusion criteria for records that do not address the outcomes raised in the review question).

Data Abstraction and Synthesis: Better to decribe: data extraction.

In my opinion, phase 3 does not correspond to the design of the systematic review. I consider that the information reported may address the way in which the results will be organized/structured, but as they describe this part (with the need to distinguish phases, it points to a more complex research design that must be redefined, for example a mixed methods).

Results:

Flowchart is the first result. Just as the authors should not structure the methodology in phases, neither should you do so with the results.

The results should not have so many citations. Could the results of phase 2 be shown in a table for better compression?

The process and establishment of the criteria for developing the categories (ID) / Themes in phase 3 is not sufficiently clarified. In Results, table 3 is not self-explanatory of the process described in the methodology of phase 3. It is not understood.

Discussion:

The discussion is scarce. I had already noted the limitations discussed in my comments during the methodology, but such limitations are not justified since they could have been solved with a better designed research protocol.

Review references according to journal standards. Include link or DOI in references that do not have.

Author Response

(The authors gave the same response as above.)

Reviewer 5 Report

Comments and Suggestions for Authors

The article contributes valuable cognitive insights into robotics in nursing, aligning with the thematic scope of the journal. However, it is not free from certain "scientific weaknesses." Below are some comments on the article:

Lines 30-38: It is true that technology plays a significant role in the broader field of medicine and can be utilized for better management of proper and effective healthcare. Nevertheless, caution must be exercised in formulating categorical opinions on this matter. Many experts worldwide emphasize that actions taken during COVID-19 were inadequate (exaggerated) in many countries, and knowledge-based models applied were flawed. On the one hand, technology can offer assistance; on the other hand, blindly following technology can lead to making erroneous decisions.

The article does not consider important literature on the subject, such as "Artificial Intelligence and Robotics in Nursing: Ethics of Caring as a Guide to Dividing Tasks Between AI and Humans," https://onlinelibrary.wiley.com/doi/10.1111/nup.12306

Lines 123-127: The authors do not explain why they include articles from January 2017 to December 2023 in their research. Were similar nursing issues not discussed earlier?

A significant research weakness: Only one database, PubMed, was utilized. In the article (under Limitations), the introduced limitations should be more explicitly and strongly justified. Simply stating that only one database was referenced and literature from 2017 to 2023 in English and Spanish was utilized does not justify the research methodologically.

Author Response

(The authors gave the same response as above.)

Round 2

Reviewer 3 Report

Comments and Suggestions for Authors

The authors did an extensive revision on the paper improvement.  Provided refinement work absolutely fit to the main requirements 

Author Response

Thank you very much for taking the time to review this manuscript and for your comments .

Reviewer 4 Report

Comments and Suggestions for Authors

Dear authors,

thank you for your respones. The changes made to the manuscript have improved the final result. I have several appreciations.

Thank you very much for adjusting the title as Scoping review. I think this is a correct change. You should adjust two aspects with this change:
1. In the material and methods section (Design) you do not need to say "Systematic" (just describe Scoping review is sufficient).
2. In this same section, with the change of review type, they can make use of the PRISMA extension for scoping review (PRISMA ScR) (Reference: Tricco, AC, Lillie, E, Zarin, W, O'Brien, KK, Colquhoun, H, Levac, D, Moher, D, Peters, MD, Horsley, T, Weeks, L, Hempel, S et al. PRISMA extension for scoping reviews (PRISMA-ScR): checklist and explanation. Ann Intern Med. 2018,169(7):467-473. doi:10.7326/M18-0850.

Correct in the abstract that the database consulted is MEDLINE to unify with the text of the manuscript (May indicate: MEDLINE (through PubMed)).

I believe there is a typo on page 3 (line 129) that corresponds to information from the previous version of the manuscript when pointing to phase 1. (In this version no phases of the revision have been pointed out).

Table 1 should not be cited twice in the text. A single citation  (see Table 1) in the text at the end of the item is better.

It is not necessary to explain in the text the use of Booleans (at the end of page 5 and the beginning of page 6). It is sufficient to describe in the text the use of the three Booleans and to cite table 2, where this content is found.

The following paragraph should be moved to the results section:

"A total of 21 records were obtained from the search strategy about nursing, robotics and competencies. A total of 88 records were obtained from the search strategy about nursing, informatics and competencies. The 109 records were reviewed by reading title and abstract. Finally 16 articles were selected for inclusion in the descriptive review".

This information is already described in the flowchart (although you can add this text in the previous explanation of the flowchart).

Reviewer 5 Report

Comments and Suggestions for Authors

I have no further questions.

Author Response

Thank you very much for taking the time to review this manuscript and for your comments.